# Learning Commonality, Divergence and Variety for Unsupervised Visible-Infrared Person Re-identification

**Jiangming Shi**[1,3]\*, **Xiangbo Yin**[2]\*, **Yachao Zhang**[2], **Zhizhong Zhang**[4,5]
**Yuan Xie**[3,4]†, **Yanyun Qu**[1,2]†
[1]Institute of Artificial Intelligence, Xiamen University
[2]School of Informatics, Xiamen University
[3]Shanghai Innovation Institute
[4]East China Normal University
[5]Shanghai Key Laboratory of Computer Software Evaluating and Testing
jiangming.shi@outlook.com  yxie@cs.ecnu.edu.cn  yyqu@xmu.edu.cn
Code: https://github.com/shijiangming1/PCLHD

## Abstract

Unsupervised visible-infrared person re-identification (USVI-ReID) aims to match specified persons in infrared images to visible images without annotations, and vice versa. USVI-ReID is a challenging yet underexplored task. Most existing methods address the USVI-ReID through cluster-based contrastive learning, which simply employs the cluster center to represent an individual. However, the cluster center primarily focuses on commonality, overlooking divergence and variety. To address the problem, we propose a Progressive Contrastive Learning with Hard and Dynamic Prototypes for USVI-ReID. In brief, we generate the hard prototype by selecting the sample with the maximum distance from the cluster center. We reveal that the inclusion of the hard prototype in contrastive loss helps to emphasize divergence. Additionally, instead of rigidly aligning query images to a specific prototype, we generate the dynamic prototype by randomly picking samples within a cluster. The dynamic prototype is used to encourage variety. Finally, we introduce a progressive learning strategy to gradually shift the model's attention towards divergence and variety, avoiding cluster deterioration. Extensive experiments conducted on the publicly available SYSU-MM01 and RegDB datasets validate the effectiveness of the proposed method.

## 1 Introduction

Visible-infrared person re-identification (VI-ReID) aims at matching the same person captured in one modality with their counterparts in another modality [1–3]. It has recently gained attention in computer vision applications like video surveillance [4] and image retrieval [5–7]. With the development of deep learning [8–11], VI-ReID has achieved remarkable advancements [12–14]. However, the development of existing VI-ReID methods is still limited due to the requirement for expensive-annotated training data [15, 16]. To mitigate the problem of annotating large-scale cross-modality data, some semi-supervised VI-ReID methods [17–19] are proposed to learn modality-invariant and identity-related discriminative representations by utilizing both labeled and unlabeled data. For this purpose, OTLA [17] proposed an optimal transport label assignment mechanism to assign pseudo-labels for unlabeled infrared images while ignoring how to calibrate noise pseudo-labels. DPIS [18] integrates two pseudo-labels generated by distinct models into a hybrid pseudo-label

---

\*Equal contribution.
†Corresponding author.

for unlabeled infrared data, but it makes the training process more complex. Although these methods have gained promising performances, they still rely on a certain number of manual-labeled data.

Several USVI-ReID methods [20–23] have proposed to tackle the issues of expensive visible-infrared annotation through contrastive learning. These methods create two modality-specific memories, one for visible features and the other for infrared features. During training, these methods consider the memory center as a prototype and minimize the contrastive loss across the features of query images and prototype. Then, these methods aggregate the corresponding prototypes based on similarity. However, the centroid prototype only stores the commonality of each person, neglecting the divergence [24–26], which causes the pseudo-labels generated by the cluster to be unreliable. Just like a normal distribution, to better reflect the data distribution of a dataset, we need not only the mean but also the variance.

In this paper, we argue that an important aspect of contrastive learning for USVI-ReID, i.e. the design of the prototype, has so far been neglected, and propose progressive contrastive learning with hard and dynamic prototype (PCLHD) method for the USVI-ReID. Firstly, we design a Hard Prototype Contrastive Learning (HPCL) to mine divergent yet meaningful information. In contrast to traditional contrastive learning methods, we choose the hard samples to serve as the hard prototype. In other words, the hard prototype is the one that is farthest from the memory center. The hard prototype encompasses distinctive information. Furthermore, we introduce the concept of Dynamic Prototype Contrastive Learning (DPCL), we randomly select samples from each cluster to serve as the dynamic prototype. DPCL effectively accounts for the intrinsic variety within clusters, enhancing the model's adaptability to varying data distributions. Early clustering results are unreliable, and utilizing hard and dynamic prototype at this stage may lead to cluster degradation. Therefore, we introduce progressive contrastive learning to gradually focus on divergence and variety.

The main contributions are summarized as follows:

- We propose a progressive contrastive learning with hard and dynamic prototype method for the USVI-ReID. We reconsider the design of prototypes in contrastive learning to ensure that the model stably captures commonality, divergence, and variety.
- We propose Hard Prototype Contrastive Learning for mining divergent yet significant information, and Dynamic Prototype Contrastive Learning for preserving the intrinsic variety in sample features.
- Experiments on SYSU-MM01 and RegDB datasets demonstrate the superiority of our method compared to existing USVI-ReID methods, and PCLHD generates higher-quality pseudo-labels than other methods.

## 2  Related Work

### 2.1  Supervised Visible-Infrared Person ReID

Visible-infrared person re-identification (VI-ReID) has drawn much attention in recent years [27–32]. Many VI-ReID methods focused on mitigating huge semantic gaps across modalities have made advanced progress, which can be classified into two primary classes based on their different aligning ways: image-level alignment and feature-level alignment. The image-level alignment methods focus on reducing cross-modality gaps by modality translation. Some GAN-based methods [33, 34] are proposed to perform style transformation for aligning cross-modality images. However, the generated images unavoidably contain noise. Therefore, X-modality [35] and its promotions [36, 37] align cross-modality images by introducing a middle modality. Mainstream feature-level alignment methods [38–40] focus on minimizing cross-modality gaps by finding a modality-shared feature space. However, the advanced performances of the above methods build on large-scale human-labeled cross-modality data, which are quite time-consuming and expensive, thus hindering the fast application of these methods in real-scenes.

### 2.2  Unsupervised Single-Modality Person ReID

The existing unsupervised single-modality person ReID methods can be roughly divided into two classes: Unsupervised domain adaption (UDA) methods, which try to leverage the knowledge transferred from labeled source domain to improve performance [41–44], and fully unsupervised

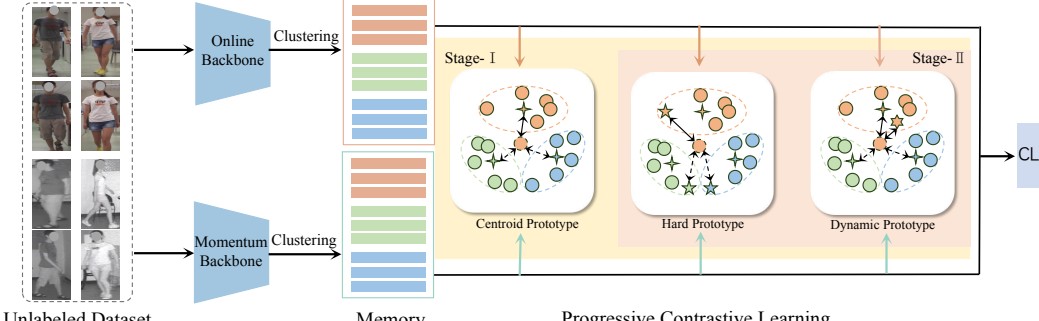

Figure 1: Framework of our PCLHD. The framework consists of two stages: the first stage employs contrastive learning with centroid prototypes to learn well-discriminative representation, and the second stage introduces contrastive learning with hard and dynamic prototypes to further focus on divergence and variety.

methods (USL), which directly train a USL-ReID model on the unlabeled target domain [21, 45]. Compared with the UDA methods, the USL methods are more challenging. Recently, cluster-contrast learning [46] has achieved impressive performance by performing contrastive learning at the cluster level. However, cluster-contrast with a uni-proxy can be biased and confusing, which fails to accurately describe the information of a cluster. To this end, the methods [47, 48] proposed maintaining multi-proxies for a cluster to adaptively capture different information within the cluster. The above methods are mainly proposed to solve the single-modality ReID task, but they are limited to solving the USL-VI-ReID task due to large cross-modality gaps.

### 2.3 Unsupervised Visible-Infrared Person ReID

Unsupervised visible-infrared person ReID (USVI-ReID) has attracted much attention due to the advantage of not relying on any data annotation. Some UDA methods [49, 17] use a well-annotated labeled source domain for pre-training to solve the USVI-ReID task. Some fully unsupervised methods [23, 22] adopt contrastive learning to boost performance, which mainly follow a two-step loop paradigm: generating pseudo-labels using the DBSCAN algorithm [50] to create memory banks with clustering centers and establishing cross-modality correspondences based on these memory banks. However, pseudo-labels are often inaccurate and rigid, CCLNet [51] leverages the text information from CLIP to afford greater semantic monitoring insights to compensate for the rigidity of pseudo-labels. Moreover, reliable cross-modality correspondences are vital to USVI-ReID, thus PGM [23] proposes a progressive graph matching framework to establish more reliable cross-modality correspondences. However, cluster centers mainly present common information while lacking distinctive information, which results in ambiguous cross-modality correspondences when meeting hard samples [52, 53].

## 3 Method

### 3.1 Problem Formulation and Overview

Given a USVI-ReID dataset $D = \{V, R\}$, where $V = \{V_i\}_{i=1}^{N_v}$ represents the visible images and $R = \{R_j\}_{j=1}^{N_r}$ denotes the infrared images. $V_i$ and $R_j$ represent the set of images corresponding to the $i$-th and $j$-th class. $N_v$ and $N_r$ denote the number of visible and infrared clusters, respectively. In the USVI-ReID task, the purpose is to train a deep neural network to obtain modality-invariant and identity-related features for matching pedestrian images with the same identity.

We propose a Progressive Contrastive Learning with Hard and Dynamic Prototype (PCLHD) method for USVI-ReID, which mainly contains online encoder, momentum encoder, and progressive contrastive learning strategy with centroid prototype, hard prototype, and dynamic prototype, as shown in Fig. 1. The online encoder is a standard network, updated through back-propagation. The momentum

encoder mirrors the structure of the online encoder, updated through the weights of the online encoder. The clustering is used to generate pseudo labels for creating cluster-aware memory, and we employ DBSCAN for clustering. PCLHD primarily focuses on representation learning, and we use PGM [23] to aggregate cross-modality memory.

## 3.2 Centroid Prototype Contrastive Learning

Following the USVI-ReID methods [22, 54], we use centroid prototype contrastive learning to optimize the online encoder in the first state, which includes memory initialization and optimization.

**Memory Initialization.** Let $\phi_0$ be the online encoder that transforms the input image to an embedding vector. At the beginning of each training epoch, all image features are clustered by DBSCAN [50] and then each cluster's representations are stored in visible memory $M_{RGB} = \{cm_1^v, cm_2^v, \cdots, cm_{N_v}^v\}$ and infrared memory $M_{IR} = \{cm_1^r, cm_2^r, \cdots, cm_{N_r}^r\}$, as follows:

$$cm_i^v = \frac{1}{|V_i|} \sum_{v \in V_i} \phi_0(v), \tag{1}$$

$$cm_j^r = \frac{1}{|R_j|} \sum_{r \in R_j} \phi_0(r), \tag{2}$$

where $|\cdot|$ denotes the number of instances belonging to specific cluster.

**Optimization.** During training, we update the two modality-specific memories by a momentum updating strategy [46]. We treat the memory center as a centroid prototype and optimize the feature extractor $\phi_0$ using contrastive learning with the centroid prototype, computed as:

$$\mathcal{L}_{CPCL}^v = \frac{1}{N_v} \sum_{i \in N_v} \frac{-1}{|V_i|} \sum_{v \in V_i} \log \frac{\exp\left(\phi_0(v) \cdot cm_i^v / \tau\right)}{\sum\limits_{j \in N_v} \exp\left(\phi_0(v) \cdot cm_j^v / \tau\right)}, \tag{3}$$

$$\mathcal{L}_{CPCL}^r = \frac{1}{N_r} \sum_{i \in N_r} \frac{-1}{|R_i|} \sum_{r \in R_i} \log \frac{\exp\left(\phi_0(r) \cdot cm_i^r / \tau\right)}{\sum\limits_{j \in N_r} \exp\left(\phi_0(r) \cdot cm_j^r / \tau\right)}, \tag{4}$$

$$\mathcal{L}_{CPCL} = \mathcal{L}_{CPCL}^v + \mathcal{L}_{CPCL}^r, \tag{5}$$

where $cm_i^{v(r)}$ is the positive centroid prototype, denoting a query and the prototype shares the same identity. The $\tau$ is a temperature hyper-parameter.

## 3.3 Hard Prototype Contrastive Learning

To ensure that the prototype effectively captures divergence within a identity, we devise a novel hard prototype for contrastive learning, which is referred to as Hard Prototype Contrastive Learning. HPCL is designed to provide a comprehensive understanding of personal characteristics, which benefits its handling of hard samples [47]. We use the online encoder $\phi_0$ to extract feature representations, and select $k$ samples that are farthest from the memory center as the hard prototype:

$$hm_i^v = \arg\max_{\forall v \in V_i} \left\| \phi_0(v) - cm_i^v \right\|, \tag{6}$$

$$hm_j^r = \arg\max_{\forall r \in R_j} \left\| \phi_0(r) - cm_j^r \right\|. \tag{7}$$

**Theorem 1.** The information entropy of hard sample prototypes is greater than the information entropy of centroid prototypes, thereby preserving greater divergence within the hard memory.

Given a set of features $\{f_1, f_2, \ldots, f_{N_c}\}$ for class $c$. The entropy $H(cm_c)$ can be approximated by the entropy of the distribution of the sample means. Considering that $cm_c$ is a convex combination of the sample features $f_i$, we have:

$$H\left(cm_c\right) = H\left(\frac{1}{N_c}\sum_{i=1}^{N_c} f_i\right). \tag{8}$$

By the convexity of entropy, we have:

$$H\left(\frac{1}{N_c}\sum_{i=1}^{N_c} f_i\right) \leq \frac{1}{N_c}\sum_{i=1}^{N_c} H(f_i). \tag{9}$$

This inequality implies that the entropy of the centroid prototype is generally lower due to the averaging effect, which reduces the divergence among the samples, leading to lower entropy. Given that $hm_c$ is the sample with the maximum individual entropy among the set $\{f_1, f_2, \ldots, f_{N_c}\}$, it follows that:

$$H(hm_c) \geq \frac{1}{N_c}\sum_{i=1}^{N_c} H(f_i) \geq H(cm_c). \tag{10}$$

Then, we construct contrastive loss with the hard prototype to minimize the distance between the query and the positive hard prototype while maximizing their discrepancy to all other cluster hard prototypes, as follows:

$$\mathcal{L}_{HPCL}^v = \frac{1}{N_v}\sum_{i\in N_v}\frac{-1}{|V_i|}\sum_{v\in V_i}\log\frac{\exp\left(\phi_0(v)\cdot hm_i^v/\tau\right)}{\sum_{j\in N_v}\exp\left(\phi_0(v)\cdot hm_j^v/\tau\right)}, \tag{11}$$

$$\mathcal{L}_{HPCL}^r = \frac{1}{N_r}\sum_{i\in N_r}\frac{-1}{|R_i|}\sum_{r\in R_i}\log\frac{\exp\left(\phi_0(r)\cdot hm_i^r/\tau\right)}{\sum_{j\in N_r}\exp\left(\phi_0(r)\cdot hm_j^r/\tau\right)}, \tag{12}$$

$$\mathcal{L}_{HPCL} = \mathcal{L}_{HPCL}^v + \mathcal{L}_{HPCL}^r, \tag{13}$$

where $hm_i^{v(r)}$ is the positive hard prototype representation and the $\tau$ is a temperature hyper-parameter.

Finally, we update the two modality-specific memories with a momentum-updating strategy:

$$hm_{i,t}^v = \alpha hm_{i,t-1}^v + (1-\alpha)\phi_0(v), \forall v \in V_i \tag{14}$$

$$hm_{i,t}^r = \alpha hm_{i,t-1}^r + (1-\alpha)\phi_0(r), \forall r \in R_i \tag{15}$$

where $\alpha$ is a momentum coefficient that controls the update speed of the memories. $t$ and $t-1$ refer to the current and last iteration, respectively.

The hard prototype contrastive learning has two main advantages: For intra-class feature learning, it ensures that the learning process does not just focus on the shared characteristics within a cluster but also considers the diverse elements, which are often more informative. For inter-class feature learning, it is also beneficial for increasing the distances between different persons. In contrast, centroid prototypes tend to average features, lacking diversity, which can affect the network's ability to extract discriminative features.

### 3.4 Dynamic Prototype Contrastive Learning

Inspired by MoCo [55] and DPM [56], we design dynamic prototype contrastive learning in order to preserve the intrinsic variety in sample features. DPCL comprises an online encoder $\phi_0$ and a momentum encoder $\phi_m$. The momentum encoder mirrors the structure of the online encoder, which is updated by the accumulated weights of the online encoder:

$$\phi_m^t = \beta\phi_m^{t-1} + (1-\beta)\phi_0^t, \tag{16}$$

where $\beta$ is a momentum coefficient that controls the update speed of the momentum encoder. $t$ and $t-1$ refer to the current and last iteration, respectively. The momentum encoder $\phi_m$ is updated by the moving averaged weights, which are resistant to sudden fluctuations or noisy updates [55].

We use the momentum encoder $\phi_m$ to extract feature representation and store them in visible memory $DM_{RGB}=\{dm_1^v, dm_2^v, \cdots, dm_{N_v}^v\}$ and infrared memory $DM_{IR}=\{dm_1^r, dm_2^r, \cdots, dm_{N_r}^r\}$. We randomly select $M$ visible/infrared samples from each cluster, denoted as $X_i^v$ and $X_j^r$.as follows:

$$F_i^v = \phi_m(X_i^v), \tag{17}$$

$$F_j^r = \phi_m(X_j^r). \tag{18}$$

We select visible dynamic prototype $dm_i^v$ from $DM_{RGB}$. In the same cluster, we select the sample farthest from the query image as the prototype. In different clusters, we choose the sample closest to the query image as the prototype:

$$dm_i^v = \begin{cases} \underset{\forall f_i^v \in F_i^v}{\arg\max} \|\phi_m(\mathrm{v}_j) - f_i^v\| & \text{if } y_j = y_i \\ \underset{\forall f_i^v \in F_i^v}{\arg\min} \|\phi_m(\mathrm{v}_j) - f_i^v\| & \text{if } y_j \neq y_i \end{cases}, \tag{19}$$

where $y_q$ and $y_i$ represent the pseudo label of the query image and the dynamic prototype, respectively. $\|\cdot\|$ denotes Euclidean norm. We obtain infrared prototype $dm_j^r$ through the same method.

The overall optimization goal of DPCL is as follows:

$$\mathcal{L}_{DPCL}^v = \frac{1}{N_v} \sum_{i \in N_v} \frac{-1}{|V_i|} \sum_{\mathrm{v} \in V_i} \log \frac{\exp\left(\phi_m(\mathrm{v}) \cdot dm_i^v/\tau\right)}{\sum\limits_{j \in N_v} \exp\left(\phi_m(\mathrm{v}) \cdot dm_j^v/\tau\right)}, \tag{20}$$

$$\mathcal{L}_{DPCL}^r = \frac{1}{N_r} \sum_{i \in N_r} \frac{-1}{|R_i|} \sum_{\mathrm{r} \in R_i} \log \frac{\exp\left(\phi_m(\mathrm{r}) \cdot dm_i^r/\tau\right)}{\sum\limits_{j \in N_r} \exp\left(\phi_m(\mathrm{r}) \cdot dm_j^r/\tau\right)}, \tag{21}$$

$$\mathcal{L}_{DPCL} = \mathcal{L}_{DPCL}^v + \mathcal{L}_{DPCL}^r, \tag{22}$$

where $dm_i^{v(r)}$ is the positive dynamic prototype representation, i.e., the query image and dynamic prototype have the same identity.

DPCL promotes a flexible and adaptable learning process, aiming to minimize discrepancies between samples and their respective dynamic prototypes, rather than rigidly aligning query images with a fixed prototype.

### 3.5 Progressive Contrastive Learning

In the initial training phases, representations are generally of lower quality. Introducing hard samples at this period could be counterproductive, potentially leading the model optimization in an incorrect direction right from the start [47, 57]. To address this issue, we introduce the Progressive Contrastive Learning, which forms the overall loss function:

$$\mathcal{L}_{PCLHD} = \begin{cases} \mathcal{L}_{CPCL}, & \text{if epoch} \leqslant E_{\text{CPCL}} \\ \lambda\mathcal{L}_{HPCL} + (1-\lambda)\mathcal{L}_{DPCL}, & \text{else} \end{cases} \tag{23}$$

where $\lambda$ is the loss weight, $E_{\text{CPCL}}$ is a hyper-parameter.

## 4  Experiment

We conduct extensive experiments to validate the superiority of our proposed method. First, we provide the detailed experiment setting, which contains datasets, evaluation protocols, and implementation details. Then, we compare our method with many state-of-the-art VI-ReID methods and conduct ablation studies. In addition, to better illustrate our method, we also exhibit further analysis. If not specified, we conduct analysis experiments on SYSU-MM01 in the all-search mode.

Table 1: Comparisons with state-of-the-art methods on SYSU-MM01 and RegDB, including SVI-ReID, SSVI-ReID and USVI-ReID methods. All methods are measured by Rank-1 (%) and mAP (%). GUR* denotes the results without camera information.

| Settings | | | SYSU-MM01 | | | | RegDB | | | |
|---|---|---|---|---|---|---|---|---|---|---|
| | | | All Search | | Indoor Search | | Visible2Thermal | | Thermal2Visible | |
| Type | Method | Venue | Rank-1 | mAP | Rank-1 | mAP | Rank-1 | mAP | Rank-1 | mAP |
| SVI-ReID | DDAG [39] | ECCV'20 | 54.8 | 53.0 | 61.0 | 68.0 | 69.4 | 63.5 | 68.1 | 61.8 |
| | AGW [58] | TPAMI'21 | 47.5 | 47.7 | 54.2 | 63.0 | 70.1 | 66.4 | 70.5 | 65.9 |
| | NFS [59] | CVPR'21 | 56.9 | 55.5 | 62.8 | 69.8 | 80.5 | 72.1 | 78.0 | 69.8 |
| | LbA [60] | ICCV'21 | 55.4 | 54.1 | 58.5 | 66.3 | 74.2 | 67.6 | 72.4 | 65.5 |
| | CAJ [1] | ICCV'21 | 69.9 | 66.9 | 76.3 | 80.4 | 85.0 | 79.1 | 84.8 | 77.8 |
| | MPANet [40] | CVPR'21 | 70.6 | 68.2 | 76.7 | 81.0 | 83.7 | 80.9 | 82.8 | 80.7 |
| | DART [27] | CVPR'22 | 68.7 | 66.3 | 72.5 | 78.2 | 83.6 | 75.7 | 82.0 | 73.8 |
| | FMCNet [38] | CVPR'22 | 66.3 | 62.5 | 68.2 | 74.1 | 89.1 | 84.4 | 88.4 | 83.9 |
| | MID [61] | AAAI'22 | 60.3 | 59.4 | 64.9 | 70.1 | 87.5 | 84.9 | 84.3 | 81.4 |
| | LUPI [62] | ECCV'22 | 71.1 | 67.6 | 82.4 | 82.7 | 88.0 | 82.7 | 86.8 | 81.3 |
| | DEEN [63] | CVPR'23 | 74.7 | 71.8 | 80.3 | 83.3 | 91.1 | 85.1 | 89.5 | 83.4 |
| | SGIEL [12] | CVPR'23 | 77.1 | 72.3 | 82.1 | 83.0 | 92.2 | 86.6 | 91.1 | 85.2 |
| | PartMix [64] | CVPR'23 | 77.8 | 74.6 | 81.5 | 84.4 | 85.7 | 82.3 | 84.9 | 82.5 |
| | CAL [65] | ICCV'23 | 74.7 | 71.7 | 79.7 | 83.7 | 94.5 | 88.7 | 93.6 | 87.6 |
| | MUN [66] | ICCV'23 | 76.2 | 73.8 | 79.4 | 82.1 | 95.2 | 87.2 | 91.9 | 85.0 |
| | SAAI [13] | ICCV'23 | 75.9 | 77.0 | 83.2 | 88.0 | 91.1 | 91.5 | 92.1 | 92.0 |
| | FDNM [67] | arXiv'24 | 77.8 | 75.1 | 87.3 | 89.1 | 95.5 | 90.0 | 94.0 | 88.7 |
| | LCNL [68] | IJCV'24 | 70.2 | 68.0 | 76.2 | 80.3 | 85.6 | 78.7 | 84.0 | 76.9 |
| SSVI-ReID | OTLA [17] | ECCV'22 | 48.2 | 43.9 | 47.4 | 56.8 | 49.9 | 41.8 | 49.6 | 42.8 |
| | TAA [19] | TIP'23 | 48.8 | 42.3 | 50.1 | 56.0 | 62.2 | 56.0 | 63.8 | 56.5 |
| | DPIS [18] | ICCV'23 | 58.4 | 55.6 | 63.0 | 70.0 | 62.3 | 53.2 | 61.5 | 52.7 |
| USVI-ReID | H2H [49] | TIP'21 | 30.2 | 29.4 | - | - | 23.8 | 18.9 | - | - |
| | OTLA [17] | ECCV'22 | 29.9 | 27.1 | 29.8 | 38.8 | 32.9 | 29.7 | 32.1 | 28.6 |
| | ADCA [20] | MM'22 | 45.5 | 42.7 | 50.6 | 59.1 | 67.2 | 64.1 | 68.5 | 63.8 |
| | NGLR [69] | MM'23 | 50.4 | 47.4 | 53.5 | 61.7 | 85.6 | 76.7 | 82.9 | 75.0 |
| | MBCCM [70] | MM'23 | 53.1 | 48.2 | 55.2 | 62.0 | 83.8 | 77.9 | 82.8 | 76.7 |
| | CCLNet [51] | MM'23 | 54.0 | 50.2 | 56.7 | 65.1 | 69.9 | 65.5 | 70.2 | 66.7 |
| | PGM [23] | CVPR'23 | 57.3 | 51.8 | 56.2 | 62.7 | 69.5 | 65.4 | 69.9 | 65.2 |
| | GUR* [22] | ICCV'23 | 61.0 | 57.0 | 64.2 | 69.5 | 73.9 | 70.2 | 75.0 | 69.9 |
| | MMM [54] | ECCV'24 | 61.6 | 57.9 | 64.4 | 70.4 | 89.7 | 80.5 | 85.8 | 77.0 |
| | **PCLHD** | **Ours** | **64.4** | **58.7** | **69.5** | **74.4** | **84.3** | **80.7** | **82.7** | **78.4** |
| | **PCLHD+MMM** | **Enhanced** | **65.9** | **61.8** | **70.3** | **74.9** | **89.6** | **83.7** | **87.0** | **80.9** |

Table 2: Ablation studies on SYSU-MM01 in all search mode and indoor search mode. "Baseline" means the model trained following PGM [23]. Rank-R accuracy(%) and mAP(%) are reported.

| | Component | | | | All Search | | Indoor Search | |
|---|---|---|---|---|---|---|---|---|
| Index | Baseline | HPCL | DPCL | PCL | Rank-1 | mAP | Rank-1 | mAP |
| 1 | ✓ | | | | 56.3 | 51.7 | 60.5 | 66.2 |
| 2 | ✓ | | ✓ | | 59.1 | 54.4 | 63.6 | 68.8 |
| 3 | ✓ | ✓ | | | 62.1 | 56.8 | 65.2 | 69.8 |
| 4 | ✓ | ✓ | ✓ | | 63.7 | 57.8 | 67.0 | 72.6 |
| 5 | ✓ | ✓ | ✓ | ✓ | 64.4 | 58.7 | 69.5 | 74.4 |

## 4.1 Experiment Setting

**Dataset.** We evaluate our method on two common benchmarks in VI-ReID: **SYSU-MM01** [71] and **RegDB** [72]. SYSU-MM01 is a large-scale public benchmark for the VI-ReID task, which contains 491 identities captured by four RGB cameras and two IR cameras in both outdoor and indoor environments. In this dataset, 22,258 RGB images and 11,909 IR images with 395 identities are collected for training. In the inference stage, the query set consists of 3,803 IR images with 96 identities and the galley set contains 301 randomly selected RGB images. RegDB is collected by an RGB camera and an IR camera, which contains 4,120 RGB images and 4,120 IR images with 412 identities. To be specific, the dataset is randomly divided into two non-overlapping sets: one set is used for training and the other is for testing.

**Evaluation Protocols.** The experiment follows the standard evaluation settings in VI-ReID, i.e., Cumulative Matching Characteristics (CMC) [73] and mean Average Precision (mAP).

**Implementation Details.** We adopt the feature extractor in AGW [58], which is initialized with ImageNet-pretrained weights to extract 2048-dimensional features. During the training stage, the

input images are resized to $288\times144$. We follow augmentations in CAJ [1] for data augmentation. In one batch, we randomly sample 16 pseudo identities, and each pseudo identity samples 16 instances. We set $M$ to be 16 for computational convenience. The number of epochs is 100, in which the first 50 epochs are trained by contrastive loss with the centroid prototype. For the last 50 epochs, we train the model by contrastive loss with both the hard and dynamic prototypes. $E_{CPCL}$ is 50. At the beginning of each epoch, we utilize the DBSCAN [50] algorithm to generate pseudo labels. During the inference stage, we use the momentum encoder $\phi_m$ to extract features and take the features of the global average pooling layer to calculate cosine similarity for retrieval. The momentum value $\alpha$ and $\beta$ is set to 0.1 and 0.999, respectively. The temperature hyper-parameter $\tau$ is set to 0.05 and the weighting hyper-parameter $\lambda$ in Eq.(23) is 0.5.

## 4.2 Results and Analysis

To comprehensively evaluate our method, we compare our method with 18 supervised VI-ReID methods, 3 semi-supervised VI-ReID methods, and 9 unsupervised VI-ReID methods. The comparison results on the SYSU-MM01 and RegDB are reported in Tab. 1.

**Comparison with USVI-ReID Methods.** As shown in Tab. 1, our method achieves superior performance compared with state-of-the-art USVI-ReID methods. MMM [54] is proposed to establish reliable cross-modality correspondences and is also the current best-performing method. Our method with MMM can achieve 65.9% in Rank-1 and 61.8% in mAP, which surpasses that of MMM by a large margin of 4.3% and 3.9%. Notably, our method even without MMM gains the best performance with 64.4% in Rank-1 and 58.7% in mAP. Although existing USVI-ReID methods mentioned in Tab. 1 have made great progress in the USVI-ReID task, the neglects of divergence and variety hinder their further improvement. They overlook divergence and variety, which often constitutes hard samples. Thus, we propose progressive contrastive learning with hard and dynamic prototypes to mine hard samples, which can guide the model to learn more robust and discriminative features.

**Comparison with SSVI-ReID Methods.** There are three SSVI-ReID methods proposed to alleviate the problem of labeling cost by using a part of annotations. Remarkably, our method achieves superior performance without any annotations, outperforming all existing SSVI-ReID methods that utilize partial annotations. Moreover, the results suggest that our method can significantly reduce the dependency on manual annotations.

**Comparison with SVI-ReID Methods.** Surprisingly, our method without annotation outperforms several SVI-ReID methods, e.g., DDAG [39], AGW [58], NFS [59], LbA [60]. This shows the immense competitiveness of PCLHD compared to SVI-ReID methods that rely on complete data annotations. The superior performance of PCLHD mainly benefits from the hard prototype and dynamic prototype contrastive learning. Additionally, we have to acknowledge that a significant disparity still exists between PCLHD and the state-of-the-art fully-supervised results.

## 4.3 Ablation Study

We conduct ablation studies on the SYSU-MM01 dataset in both all-search and indoor-search modes to show the effectiveness of each component in our method. The results are shown in Tab. 2.

**Baseline Settings.** We use PGM [23] as our baseline. Although PGM has achieved a promising performance on the USVI-ReID task, the neglect of hard samples hinders its further improvement.

**Effectiveness of HPCL.** The HPCL is proposed to mine divergence. As shown in Tab. 2, When adding the HPCL on Baseline, the performance improves a large margin of 5.8% in Rank-1 and 5.1% in mAP, respectively. It shows that divergence can be effectively mined using hard prototype contrast learning, facilitating the model to learn more discriminative features.

**Effectiveness of DPCL.** The DPCL is proposed to mine variety. The results show that Rank-1 accuracy can be improved by 2.8% in Rank-1 and 2.7% in mAP when adding the DPCL on Baseline, which confirms that contrastive learning with dynamic prototype can learn variety.

**Effectiveness of PCL.** PCL is introduced to smoothly shift the model's attention from commonality to divergence and variety. The results show that Rank-1 accuracy can be improved by about 1% in Rank-1 and mAP compared to adding simultaneously the HPCL and DPCL on the Baseline. This confirms that progressive contrastive learning plays a valuable role in assisting HPCL and DPCL.

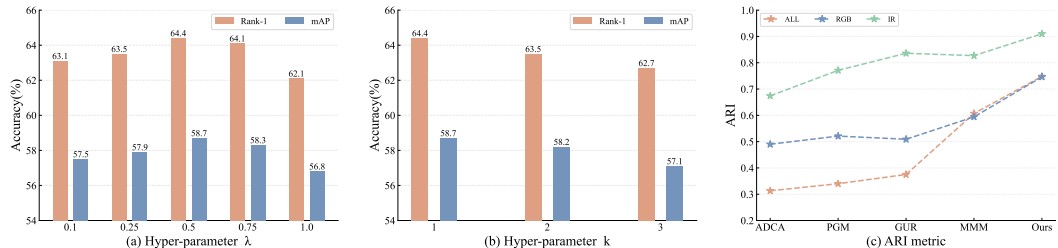

Figure 2: (a) The effect of hyper-parameter $\lambda$ with different values. (b) The effect of hyper-parameter $k$ with different values. (c) Comparisons with ARI values of different methods.

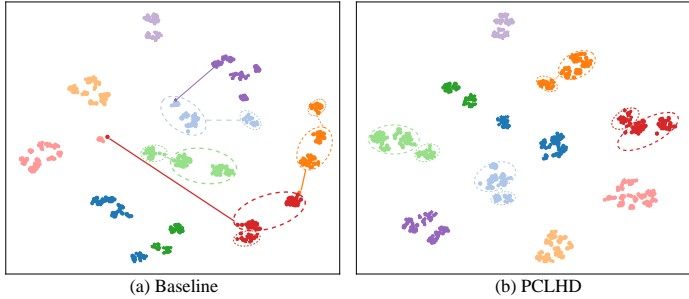

Figure 3: The t-SNE visualization of 10 randomly selected identities. Different color indicates different IDs. Circle means visible features and the pentagram means infrared features.

Surprisingly, contrastive learning with both hard and dynamic prototypes significantly exceeds the baseline by a large margin of 8.1% in Rank-1 and 7.0% in mAP. The HPCL and DPCL can complement each other to learn divergence and variety, which effectively guides the network to learn more robust and discriminative features.

## 4.4 Further Analysis

**Hyper-parameter analysis.** Hyper-parameter $\lambda$ is a weighting parameter to trade-off $L_{HPCL}$ and $L_{DPCL}$. Fig. 2 (a) presents the results under different values of $\lambda$. We can observe that when $\lambda$ is small, i.e., $L_{DPCL}$ contributes more to the model, the performance degrades. However, when $\lambda$ is large, i.e., $L_{HPCL}$ contributes heavily to the model, the model both achieves superior performance. Note that when $\lambda = 1$, i.e., the proposed method is trained without DPCL, the performance drops significantly. $\lambda$ is finally set to 0.5 and our method achieves the best performance of 64.4% in Rank-1. Moreover, we also analyze the effect of the number of hard samples at hard prototype. As shown in Fig. 2 (b), we vary the $k$ from 1 to 3 and keep the other hyper-parameters fixed, which shows that PCLHD achieves the best performance when $k = 1$. Hard samples are distributed in multiple directions, so multiple hard samples cannot be represented by a single prototype. This is why using more hard samples as prototypes leads to a decline in overall performance

**The ARI metric.** Following MMM [54], we utilize the Adjusted Rand Index (ARI) metric for clustering evaluation. The larger the ARI value, the higher the clustering quality. In Fig. 2 (c), "RGB" and "IR" denote the ARI values of visible and infrared clusterings, which can measure the quality of visible and infrared pseudo-labels. "ALL" means the ARI values of overall clusterings, which can evaluate the reliability of cross-modality correspondences. PCLHD surpasses other methods significantly on all of the mentioned ARI values, which demonstrates PCLHD can effectively mine divergence and variety to improve clustering quality.

**Visualization.** As shown in Fig. 3, we visualize the t-SNE map of 10 randomly chosen identities from SYSU-MM01. Compared to the baseline, the distribution of the same identity from the same modality is more compact and the distance of the same identity from different modalities is closer together. Moreover, some hard samples in the baseline are incorrectly clustered, while these hard samples are well clustered in our PCLHD, which shows the effectiveness of the proposed PCLHD.

# 5 Conclusion and Limitation

In this paper, we propose a novel method for USVI-ReID called Progressive Contrastive Learning with hard and dynamic prototype (PCLHD), which learns commonality, divergence and variety. To be specific, we design Hard Prototype Contrastive Learning to mine divergent yet significant information and Dynamic Prototype Contrastive Learning to preserve intrinsic variety features. Furthermore, we introduce a progressive learning strategy to incorporate both HPCL and DPCL into the model. Extensive experiments demonstrate that PCLHD outperforms state-of-the-art USVI-ReID methods.

This work relies on DBSCAN to generate pseudo-labels. However, for extremely large-scale datasets, DBSCAN's performance may be limited, which could affect the overall effectiveness of our approach. To address the limitation, we plan to explore hierarchical clustering in future research to better handle large-scale datasets.

## Broader Impacts

This work was developed using publicly available datasets and aims to enhance the capabilities of VI-ReID, which plays a vital role in scenarios where traditional ReID systems fail, such as in low-light or nighttime conditions. VI-ReID offers significant benefits in improving security and surveillance by enabling more reliable identification across varying environmental conditions. Importantly, this work raises no ethical, safety, or environmental concerns, and no harm was inflicted on living beings during the research. However, we acknowledge the risk of misuse, particularly privacy invasion if used to track individuals in public spaces without appropriate regulation. While VI-ReID does not directly identify specific individuals, its unauthorized deployment could still result in significant privacy violations. Therefore, public surveillance systems using VI-ReID should be controlled by authorized entities, ensuring proper regulatory frameworks and adherence to ethical standards.

## Acknowledgements

This work was supported by the National Natural Science Foundation of China (No. 62176224, 62176092, 62222602, 62306165, 62106075, 62476090), Natural Science Foundation of Shanghai (23ZR1420400), Natural Science Foundation of Chongqing (CSTB2023NSCQ-JQX0007), China Computer Federation (CCF) Lenovo Blue Ocean Research Fund, China Academy of Railway Sciences No. 2023YJ357.

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
