# OpenReview forum: "Learning Commonality, Divergence and Variety for Unsupervised Visible-Infrared Person Re-identification"
_NeurIPS.cc/2024/Conference — NeurIPS 2024 poster_

### Official Review · Reviewer_n3oM · 2024-07-10

**Soundness:** 4
**Presentation:** 3
**Contribution:** 3
**Rating:** 7
**Confidence:** 5

**Summary:**

This paper proposed a Progressive Contrastive Learning with Multi-Prototype method for USVI-ReID, which learns both shared and diverse information. The Hard Prototype Contrastive Learning method aims to mine distinctive yet significant information, while the Dynamic Prototype Contrastive Learning method is designed to preserve intrinsic diversity features. The experimental results significantly outperform existing methods, validating the effectiveness of the proposed method.

**Strengths:**

1) The paper proposes a straightforward yet effective method, and the experimental results are promising for the USVI-ReID task.
2) The paper is well-written and easy to follow.
3) The proposed method is well-motivated. Its prototype designs enhance the learning of commonality, divergence, and variety. It makes up for the limits of traditional center prototype-based approaches.
4) Figure 2(c) demonstrates that the proposed method generates higher-quality pseudo labels compared to other state-of-the-art methods.

**Weaknesses:**

1) From Table 2, it is evident that the method is based on a relatively strong baseline. Does the paper use different baselines, like ADCA, still achieve good results?
2) The paper introduces a momentum backbone in the PCLHD method but does not fully explain its advantages.
3) Cross-modality associations are especially important for USVI-ReID, what is the association strategy used in this paper? Some details need to be supplemented for clarity.

**Questions:**

What insights does this method offer for other multimodal tasks? How can the approach be adapted to enhance performance in different multimodal scenarios?

**Limitations:**

Although the method shows significant performance improvement over other unsupervised methods, there is still a considerable gap when compared to fully supervised methods.

---

> ### Author Rebuttal · Authors · 2024-08-06
>
> ### Weaknesses
>
> **W1: From Table 2, it is evident that the method is based on a relatively strong baseline. Does the paper use different baselines, like ADCA, still achieve good results?**
>
> A1: We acknowledge the importance of evaluating our method with different baselines. As shown in following Table, we have included results with two additional baselines, ADCA and MMM. Our method, when combined with these baselines, still achieves significant improvements. For instance, ADCA's Rank-1 accuracy improves from 45.5% to 52.5% and mAP from 42.7% to 48.5% when integrated with our PCLHD. Similarly, MMM's Rank-1 accuracy improves from 61.6% to 65.9% and mAP from 57.9% to 61.8%, and the gain is remarkable by 4.3% and 3.9%. These results demonstrate the effectiveness of our approach across different baselines.
> |  Methods   | Rank-1@All | mAP@All | Rank-1@Indoor | mAP@Indoor |
> | :--------: | :--------: | :--------: | :-----------: | :-----------: |
> |    ADCA    |    45.5    |    42.7    |     50.6      |     59.1      |
> | ADCA+PCLHD |    52.5    |    48.5    |     56.6      |     62.3      |
> |    MMM     |    61.6    |    57.9    |     64.4      |     70.4      |
> | ADCA+PCLHD |    65.9    |    61.8    |     70.3      |     74.9      |
>
> **W2: The paper introduces a momentum backbone in the PCLHD method but does not fully explain its advantages.**
>
> A2: In unsupervised learning, noise is inevitable, and dynamic prototypes can introduce a certain degree of instability. The momentum encoder mirrors the structure of the online encoder and is updated by the accumulated weights of the online encoder. Our method is resistant to sudden fluctuations or noisy updates, leading to more consistent and robust feature representations.
>
> **W3: Cross-modality associations are especially important for USVI-ReID, what is the association strategy used in this paper? Some details need to be supplemented for clarity.**
>
> A3: In USVI-ReID methods, the process is typically divided into two stages. The first stage involves intra-modality identity alignment learning, while the second stage focuses on cross-modality associations. Our strategy emphasizes obtaining accurate intra-modality alignment suitable for the second stage. As shown in Figure 3c, our cross-modality association accuracy is significantly higher than existing unsupervised methods. We utilize progressive contrastive learning, including centroid and hard prototype contrastive learning, to achieve this. For inter-modality association, we build on the well-aligned intra-modality identities and use existing methods like PGM and MMM to handle the association. This ensures effective cross-modality matching by leveraging the strengths of both intra- and inter-modality learning.
>
> ### Questions
>
> **Q1: What insights does this method offer for other multimodal tasks? How can the approach be adapted to enhance performance in different multimodal scenarios?**
>
> A1: Our method offers valuable insights for multimodal tasks by demonstrating the effectiveness of progressive contrastive learning with hard and dynamic prototypes. When dealing with multimodal feature alignment and representation, it may be beneficial to make the intra-modality  feature alignment and then make the inter-modality feature alignment. By mining divergence and variety through these prototypes, our method enhances clustering quality and feature discrimination, which is crucial for aligning cross-modal features. For instance, in image-text matching tasks, hard and dynamic prototypes help capture divergence and variety in visual and textual data, leading to better alignment between the two modalities.

---

> > ### Comment · Reviewer_n3oM · 2024-08-09
> >
> > Thank you very much for addressing most of my concerns. Considering the other reviews, I believe the revised manuscript will be nice, and I am willing to raise my score.
> >
> > However, after re-reading the manuscript, I have few new questions and suggestions:
> >
> > (1) Do I understand correctly that you do not plan to include a limitations section? If that is the case, I strongly urge you to consider adding such a section.
> >
> > (2) In Eq. 19, Does $v_j$  refer to the query image? Could you please clarify this here? Besides, $y_q$ should be $y_j$ in line 180.
> >
> > (3) Please note that there is a typo in line 175.

---

> ### Author Response · Authors · 2024-08-10
>
> Thank you so much for acknowledging the strengths of our method again. We have carefully considered your constructive comments and here are the answers to your concerns.
>
> **C1: Do I understand correctly that you do not plan to include a limitation section? If that is the case, I strongly urge you to consider adding such a section.**
>
> A1: We greatly value the reviewers' feedback and will be adding a limitation section as shown in the general response to the manuscript. We believe this addition will provide a more complete understanding of our work.
>
> **C2: In Eq. 19, Does $v_j$ refer to the query image? Could you please clarify this here? Besides, $v_q$ should be $v_j$ in line 180.**
>
> A2: Thank you for your detailed feedback. in Eq. 19, $v_j$ denotes the query image. Besides, We will also correc $y_q$ to$ y_j$ in line 180.
>
> **C3: Please note that there is a typo in line 175.**
>
> A3: Thank you for pointing out the typo. We will thoroughly proofread the manuscript and correct typos to the best of our efforts.

---

> > ### Comment · Reviewer_n3oM · 2024-08-13
> >
> > Thank you for the response, which addressed my concerns.

---

### Official Review · Reviewer_i5Cg · 2024-07-11

**Soundness:** 3
**Presentation:** 4
**Contribution:** 3
**Rating:** 7
**Confidence:** 4

**Summary:**

This paper proposes a method for unsupervised visible-infrared person re-identification called Progressive Contrastive Learning with Hard and Dynamic Prototype. This method focuses on capturing commonality, divergence, and variety. By introducing hard prototype contrastive learning to emphasize divergence and dynamic prototype contrastive learning to preserve intrinsic feature variety, the approach addresses limitations of existing methods. The proposed progressive learning strategy further enhances model robustness and prevents cluster deterioration. Experimental results on two benchmark datasets, SYSU-MM01 and RegDB, show significant improvements in performance, with notable increases in mAP and Rank-1 accuracy compared to the current state-of-the-art methods.

**Strengths:**

1.The discussion on the limitations of existing methods and how the proposed method addresses these limitations is compelling. The motivation for focusing on commonality, divergence, and variety is well-articulated.
2.The methodology is detailed and well-structured. The explanation of hard prototype contrastive learning, and dynamic prototype contrastive learning is clear.
3.The comparison with state-of-the-art methods is comprehensive. The proposed method’s superior performance.
4.The further analysis, including ARI metric, provides additional insights into the robustness and effectiveness of the method. The analysis of the results is insightful, explaining why the proposed method outperforms others.

**Weaknesses:**

1.While introducing hard prototype contrastive learning can capture divergences, selecting hard samples may lead to model instability during the early training stages.
2.Although a progressive contrastive learning strategy is introduced to mitigate the above problem, balancing the impact of hard samples remains a challenge.
3.The strategy for selecting dynamic prototypes might require more complex implementation and higher computational resources on larger datasets.
4.Although the paper mentions that the code will be released after acceptance, it currently lacks detailed instructions for reproducibility.
5.The accuracy of pseudo-labels directly impacts the model training. Some parameters in DBSCAN are very important and should be analyzed further.

**Questions:**

1.Is L_CPCL involved in the optimization process during the second stage?
2.Is the comparison fair with GUR without camera information?

**Limitations:**

The method relies on the DBSCAN clustering algorithm to generate pseudo-labels. For large-scale datasets, DBSCAN might not perform well, thereby affecting the overall method's performance.

---

> ### Author Rebuttal · Authors · 2024-08-06
>
> ### Weaknesses
>
> **W1: While introducing hard prototype contrastive learning can capture divergences, selecting hard samples may lead to model instability during the early training stages. Although progressive contrastive learning strategy is introduced to mitigate the above problem, balancing the impact of hard samples remains a challenge.**
>
> A1: Our ablation experiments show that even starting with HPCL and DPCL without using PCL, the method achieves the Rank-1 accuracy of 63.7%, only with the 0.7% decrease. This is because CPCL is involved in training, helping the model maintain stability when learning divergence and variety. Additionally, our experiments demonstrate that method is not highly sensitive to the $E_{CPCL}$ hyperparameter. For more details, please refer to Q3 for Reviewer mgXA.
>
> **W2: The strategy for selecting dynamic prototypes might require more complex implementation and higher computational resources on larger datasets.**
>
> A2: We acknowledge that dynamic prototype learning requires more complex implementation and higher computational resources during training. However, it is important to note that during the inference stage, this process is not needed. The complexity and computational requirements are the same as other unsupervised methods, with no additional overhead introduced.
>
> **W3: Although the paper mentions that the code will be released after acceptance, it currently lacks detailed instructions for reproducibility. The accuracy of pseudo-labels directly impacts the model training. Some parameters in DBSCAN are very important and should be analyzed further.**
>
> A3: For fair comparisons, we follow the experimental details provided by ADCA and PGM for all other aspects. The detailed parameters are as follows: We use Adam optimizer to train the model with weight decay 5e-4. The initial learning rate is 3.5e-3 and decays 10 times every 20 epochs. The maximum distance for DBSCAN is set to 0.6 on SYSU-MM01 and 0.3 on RegDB. The minimal number for two datasets is set to 4 during clustering. We will add these details to the supplementary materials to enhance reproducibility.
>
> ### Questions
>
> **Q1: Is $L_{CPCL}$ involved in the optimization process during the second stage?**
>
> A1: $L_{CPCL}$ does participate in the optimization of the second stage. As indicated by the background color in Figure 2, $L_{CPCL}$ covers both stages.
>
> **Q2: Is the comparison fair with GUR without camera information?**
>
> A2: GUR shows two results, with and without camera information, with the latter being lower. All the unsupervised methods in Table 1, including ours, do not use camera information. To ensure a fair comparison with all methods, we present the results of GUR without camera information, not with the intention of diminishing GUR’s results. GUR vs. our method achieves 63.5% vs. 65.9% Rank-1 accuracy with camera information, and our method outperforms GUR’s best result by 2.4%. Moreover, GUR vs. our method achieves 73.9% vs. 89.6% Rank-1 accuracy on RegDB (V2T), resulting in a remarkable gain of 15.7%.

---

> > ### Comment · Reviewer_i5Cg · 2024-08-13
> >
> > After reviewing the rebuttal carefully, the authors have addressed my previous concerns in detail. I have also considered the opinions of other reviewers:\
> > (1)I agree with the other reviewers that the authors should add the limitations discussed in the general rebuttal to the original manuscript.\
> > (2)I agree more with Ethics Reviewer fgoD than with Ethics Reviewer zWHs. I argue this work is largely technical and does not present direct ethical concerns.\
> > Considering the opinions of other reviewers, I have decided to increase the score from 7 to 8.

---

> > > ### Author Response · Authors · 2024-08-13
> > >
> > > We sincerely appreciate your response and for your willingness to improve the score.

---

> ### Author Response · Authors · 2024-08-12
>
> Dear Reviewers,\
> Thanks very much for your time and valuable comments. We understand you're busy. But as the window for responding and paper revision is closing, would you mind checking our response and confirm whether you have any further questions? We are very glad to provide answers and revisions to your further questions.\
> Best regards and thanks,\
> Authors of # 1045

---

### Official Review · Reviewer_mgXA · 2024-07-12

**Soundness:** 3
**Presentation:** 2
**Contribution:** 3
**Rating:** 4
**Confidence:** 4

**Summary:**

The paper is presented in the field of unsupervised visible-infrared person re-identification (USVI-ReID) which aims to match individuals in infrared images to those in visible images without annotations. Existing methods often use cluster-based contrastive learning, which fails to account for divergencies by focusing solely on commonalities. To overcome this, the authors propose a Progressive Contrastive Learning method with Hard and Dynamic Prototypes, which emphasizes divergencies and feature variety through the use of hard prototypes and a progressive learning strategy.

**Strengths:**

The motivation behind the proposed methodology is exceptionally clear, as is the well-structured solution scheme. The logic of the presentation is sound, and the progression of the proposed solution is easy to follow, making the entire approach both coherent and engaging.

The innovative idea of adopting contrastive learning to enhance discrimination by incorporating diversity information is particularly noteworthy. This approach stands out as it not only addresses existing challenges but also brings a fresh perspective to the field.

The results, which have been demonstrated on two benchmark datasets, are truly impressive. They outperform recent state-of-the-art solutions that have been presented in top-tier venues in the computer vision (CV) field over the past few years. This achievement highlights the effectiveness and robustness of the proposed methodology.

**Weaknesses:**

The concept of farthest equal diversity in the HPCL (equations 6 and 7) is based on a strong assumption. Distance is a weak measure and a poor indicator of how the samples within a cluster are actually spread out. While variance is a simple metric, it captures the statistical distribution of the samples within the cluster more effectively.

The DPCL section is not clearly articulated. Specifically, the concept of the query and its application in equation 19 are obscure and require further clarification.

The loss function L_PCLHD  relies on the constant E_CPCL , which is arbitrarily set to 50 without any justification or ablation study to support this choice. From the results presented, it appears that all the ablation studies commence with 50 epochs of CPCL.

Additionally, although it is common practice to integrate the proposed methodology into unsupervised solutions, I believe that knowing the number of classes in advance and setting the number of clusters accordingly constitutes a form of supervision. While it is less intensive than labeling every sample with its respective class, it is still a form of supervision. In my opinion, these methods should be categorized under "unlabeled samples" rather than "unsupervised" learning.

**Questions:**

Did the authors explore alternative methods for representing diversity aside from using the farthest sample? If such exploration was undertaken, it would be beneficial to discuss the findings. Conversely, if no alternative methods were considered, does this represent a limitation of the proposed solution? Understanding the rationale behind the chosen approach or the constraints of the methodology would add valuable context to the study.

How does the performance of the proposed solution vary with different values of the E_CPCL constant? Was any ablation study conducted to investigate this aspect? An examination of the impact of varying E_CPCL  on the solution's effectiveness would provide deeper insights into the robustness and adaptability of the proposed method. Such analysis would be crucial for validating the consistency and reliability of the results.

**Limitations:**

The limitations of the proposed approach are not adequately addressed. The weaknesses related to assumptions (e.g. diversity) and constant values definitions, should have brought the authors to a deeper ablation study and thus comprehension of the limitations of the proposed solution.

---

> ### Author Rebuttal · Authors · 2024-08-06
>
> **Q1: The use of the farthest sample as a method for representing diversity in the HPCL (as described in Equations 6 and 7) relies on the assumption that distance is an adequate measure of sample spread within a cluster. Given that distance can be a weak indicator and variance might offer a better representation of statistical distribution, did the authors explore alternative methods for representing diversity, such as variance or other metrics? If so, please discuss the findings.**
>
> A1: We chose the farthest sample distance measure to represent diversity due to its simplicity and effectiveness. Distance measure is a common approach in machine learning and person re-identification, particularly in subspace learning. Thank you for your suggestion, we also explored using variance as an alternative measure for diversity. Our experiments indicate that using variance resulted in 2.8% reduction in Rank-1 compared to the our HPCL, as shown in following table. The reason for this is that our HPCL focuses on the extreme points in the feature space, which helps the model focus on the divergent samples, rather than simply minimizing intra-class variance.
> |  Metrics   | Rank-1@All | mAP@All | Rank-1@Indoor | mAP@Indoor |
> | :--------: | :--------: | :--------: | :-----------: | :-----------: |
> |    Variance     |    61.6   |    55.8    |     65.2      |     70.9      |
> | HPCL |    64.4    |    58.7    |     65.9      |     74.4      |
>
>
> **Q2: The DPCL section is not clearly articulated. Specifically, the concept of the query and its application in equation 19 are obscure and require further clarification.**
>
> A2: We apologize for the lack of clarity in the DPCL section. The query in this context refers to the feature representation of the input image that is being compared to the dynamic prototype. We followed the definition of the query as used in MoCo, PGM, and GUR. We will revise the manuscript to make this explanation clearer.
>
> **Q3: How does the performance of the proposed solution vary with different values of the E_CPCL constant? Was any ablation study conducted to investigate this aspect? An examination of the impact of varying E_CPCL on the solution's effectiveness would provide deeper insights into the robustness and adaptability of the proposed method. Such analysis would be crucial for validating the consistency and reliability of the results.**
>
> A3: We understand the concern regarding the constant $E_{CPCL}$ being set to 50. We have now conducted ablation studies to justify this choice. Our experiments tested various values for $E_{CPCL}$, and we found that 50 is not the best choice. The results showed that 40 epochs provided the best performance. We will include these ablation study results in the revised manuscript to support our choice of $E_{CPCL}$.
>
> | $E_{CPCL}$ | Rank-1@All | mAP@All | Rank-1@Indoor | mAP@Indoor |
> | :----: | :--------: | :-----: | :-----------: | :--------: |
> |   10   |    60.3    |  56.2   |     65.1      |    70.1    |
> |   20   |    62.0    |  57.5   |     66.7      |    71.6    |
> |   30   |    64.2    |  59.4   |     68.4      |    72.7    |
> |   40   |    64.6    |  59.8   |     70.0      |    75.0    |
> |   50   |    64.4    |  58.7   |     69.5      |    74.4    |
> |   60   |    64.0    |  59.5   |     68.7      |    73.2    |
> |   70   |    62.8    |  58.5   |     68.8      |    73.0    |
>
> **Q4: Although it is common practice to integrate the proposed methodology into unsupervised solutions, I believe that knowing the number of classes in advance and setting the number of clusters accordingly constitutes a form of supervision. While it is less intensive than labeling every sample with its respective class, it is still a form of supervision. In my opinion, these methods should be categorized under "unlabeled samples" rather than "unsupervised" learning.**
>
> A4: We appreciate your insight regarding the classification of our method. However, our pseudo-labels relies on the DBSCAN clustering algorithm, which does not require prior knowledge of the number of classes. For instance, while the actual number of classes in the dataset is 395, our clustering results in 384 clusters. This slight variation demonstrates that our method can effectively approximate the true number of classes without supervision.

---

> > ### Comment · Reviewer_mgXA · 2024-08-09
> >
> > The authors have successfully replied to some of my concerns. I still have doubts about using distance in HPCL since in many Re-ID solution there is significant effort also in metric learning showing that distance is not capturing all the characteristics of the feature space. Anyhow the authors show that simple distance is better than variance. I appreciate the effort.
> > Concerning the E_{CPCL} the authors confirm the necessity of ablation on such a parameters since 40 is a better value. What is missing is a comment on why such a parameters has such an impact and with such characteristics.
> > With the rebuttal I would like to raise my score to borderline accept.

---

> ### Author Response · Authors · 2024-08-10
>
> Thank you once again for providing feedback on our paper and for your willingness to consider improving the score.
>
> **C1: I still have doubts about using distance in HPCL since in many Re-ID solution there is significant effort also in metric learning showing that distance is not capturing all the characteristics of the feature space.**\
> A1: We agree with your observation that distance is not a perfect method for capturing all the characteristics of the feature space. **However, the key idea of HPCL is to focus the network on divergence using distance-based hard prototype learning, rather than on measurement methods.** We have used the simple distance measure to verify that focusing on divergence is important. We also acknowledge that there are more complex ways to measure divergence, and we believe that using more advanced metrics could improve our approach.
>
> **C2: Concerning the $E_{CPCL}$ the authors confirm the necessity of ablation on such a parameters since 40 is a better value. What is missing is a comment on why such a parameters has such an impact and with such characteristics.**\
> A2: The parameter $E_{CPCL}$ is crucial in determining the timing for focusing on divergence and variety. When $E_{CPCL}$ is set to 20, the hard prototypes are primarily composed of noisy samples. However, when $E_{CPCL}$ is set to 40, these prototypes are more likely to consist of clear hard samples. For instance, at the 20th epoch, the ARI score for IR image pseudo-label accuracy is only 0.68, but it improves to 0.89 at the 40th epoch, which demonstrates how $E_{CPCL}$ affects the quality and composition of prototypes. When $E_{CPCL}$ is set beyond 40, the network has already somewhat converged, which reduces the impact of proposed method.

---

> > ### Author Response · Authors · 2024-08-13
> >
> > Dear Reviewer mgXA,
> >
> > Thank you for your previous feedback and for mentioning your intention to raise the score. We noticed that the score has not been updated. As the discussion phase comes to an end, we want to ensure that we have thoroughly addressed all of your concerns. If there are any additional points of concern that you believe would contribute to the improvement of our manuscript, please let us know. We genuinely appreciate your expertise and the dedicated time you have devoted to reviewing our work.
> >
> > Warmest regards,
> >
> > Authors

---

### Official Review · Reviewer_edeS · 2024-07-15

**Soundness:** 3
**Presentation:** 2
**Contribution:** 3
**Rating:** 5
**Confidence:** 4

**Summary:**

The paper proposes a method for unsupervised visible-infrared person ReID. Towards this goal, it extends the PGM method with a progressive contrastive learning with hard and dynamic prototypes. Initially a typical centroid prototype contrastive learning approach is used. A mixed batch of images is encoded through an online encoder and contrastive learning is used to update this encoder using contrastive learning with the centroid prototypes computed by DBSCAN clustering the training data and updating it with a momentum-updating strategy throughout the training. The paper hypothesizes that additionally hard samples should be used as prototypes to find divergent samples and dynamic prototypes should be used to preserve variability during training. Hard prototypes are selected as those samples within a cluster that are farthest away from the centroid prototypes and dynamic prototypes are random samples from a cluster and encodes them with an encoder that keeps a momentum version of the online weights. Since the hard and dynamic prototypes might not be well-defined at the start of training, the proposed method initially uses the standard centroid prototype contrastive learning and after a certain period of training, changes to a loss function based on hard and dynamic prototype contrastive learning. This setup shows significant improvements of the PGM baseline, but also over the current state of the art.

**Strengths:**

- The paper is fairly easy to follow.
- The results look strong.
- Code is promised to be released upon acceptance.
- Overall the idea makes sense and the ablations do show that the different parts of the method are relevant.

**Weaknesses:**

- L215 states that each batch contains 16 person IDs and 16 samples per person. Is this a confusion and it should be based on the assignments from the clustering? Or are these actually person IDs from the annotations? This is kind of a big point, so please make sure to answer this in the rebuttal. I checked some other papers and there similar things are mentioned, but if this is truly based on annotations, the approach CANNOT be called unsupervised, unless you show you can do without annotations here!

- The novelty isn't that great, the base method PGM covers most of the approach and the real novelty lies in a slight modification to the contrastive learning setup for a part of the training. The dynamic prototype learning, as the paper states, is inspired by MoCo and when searching a bit for hard examples in contrastive learning, there is also quite some literature available. Nevertheless, in this specific setting, there is a contribution of showing the improvements are possible.
- I find the related work very weak! The paper is completely focused on the contrastive learning aspect of the PGM method, however, the related work section is purely focused on Visible-Infrared and unsupervised person ReID. Why is there no discussion about unsupervised learning in general, specifically with a focus on contrastive learning and its modifications?!
- While the performance boost in general is quite good, figure 2a does raise the question how stable it is w.r.t to some of the hyperparameters. Then again a fixed value of 0.5 isn't too unexpected and I guess it might have even been an initial default value. Nevertheless an actual validation set for these kinds of ablations would have been valuable, but I guess also uncommon in the community.
- Overall the discussion of the results is often a bit meaningless and just describes the values in the tables, partially even in a bit of a weird fashion. For example L238-240 points out that the proposed method "remarkably" is able to outperform even some semi-supervised methods, but quite some other unsupervised methods in the table also do so. Then in L241-246 it's pointed out that even some supervised methods from 2021 are outperformed, but almost every other supervised method outperforms every single unsupervised method in the table.

**Questions:**

- Why are the "Baseline" results in table 2 so different from the PGM results? I get the fluctuation of ~1% for All Search, but why is Indoor Search so much better in your training setup?
- Why did you duplicate almost every single equation just for visible and infrared? This is a major waste of space and could have been used for actually relevant discussions.
- In figure 2c, it very much confused me to see a line between the data points since the different methods are in no way connected.
- Is figure 3 generated based on training of test images?

**Limitations:**

The authors claim "We have not found the limitations of the work." but then again the paper is far from solving the problem. Given that this indicates there are obviously limitations, I find it somewhat lazy to not put some effort into coming up with a meaningful discussion of what still needs to be improved.

And then this is actually outrageous: "There is no societal impact of the work performed." The complete paper is about person re-identification, now even beyond the visible spectrum. Performing this kind of research can have major effects on how effective surveillance is and it's ridiculous to state that there is no impact! I will not judge someone for doing this research out of scientific curiosity, but thinking it has no impact and propagating that belief is not acceptable.

---

> ### Author Rebuttal · Authors · 2024-08-06
>
> ### Weaknesses
> **W1: L215 states that each batch contains 16 person IDs and 16 samples per person. Is this a confusion and it should be based on the assignments from the clustering? Or are these actually person IDs from the annotations?**
>
> A1: We apologize for the confusion. The statement in L215 indeed refers to the selection of samples based on the assignments **from the clustering, not annotations.** We followed the same experimental setup as existing unsupervised VI-ReID methods such as ADCA, PGM, and GUR. We will clarify this in the revised version.
>
> **W2: The novelty isn't that great, the base method PGM covers most of the approach and the real novelty lies in a slight modification to the contrastive learning setup for a part of the training.**
>
> A2: PGM solves the USL-VI-ReID by creating modality-specific memories and minimizing contrastive loss across the query images and centroid prototypes. The limitation of PGM is that it only focuses on the commonality among features,  which causes the pseudo-labels generated by the clusters to be unreliable, as shown in Figure 2c. Just like a normal distribution, which requires both the mean and the variance to accurately reflect the data distribution, HPCL and DPCL are designed to focus both divergence and variety. Importantly, we address issues not only present in PGM but also in most unsupervised VI-ReID methods, which overlook aspects of divergence and variety. As shown in the following table, our method significantly improves the performance of other methods. Specifically, when combined with PCLHD, MMM's Rank-1 improves from 61.6% to 65.9%. Additionally, we need to correct a misunderstanding: our momentum update is inspired by MoCo, not dynamic prototype learning. MoCo focuses on contrastive learning using a queue and momentum encoder. It does not involve any dynamic prototype learning.
> |  Methods   | Rank-1@All | mAP@All | Rank-1@Indoor | mAP@Indoor |
> | :--------: | :--------: | :--------: | :-----------: | :-----------: |
> |    ADCA    |    45.5    |    42.7    |     50.6      |     59.1      |
> | ADCA+PCLHD |    52.5    |    48.5    |     56.6      |     62.3      |
> |    MMM     |    61.6    |    57.9    |     64.4      |     70.4      |
> | ADCA+PCLHD |    65.9    |    61.8    |     70.3      |     74.9      |
>
> **W3: Why is there no discussion about unsupervised learning in general, specifically with a focus on contrastive learning and its modifications?**
>
> A3: We agree that the related work can be strengthened to include more discussion on unsupervised learning in general. We will revise the related work in the future version.
>
> **W4: While the performance boost in general is quite good, figure 2a does raise the question how stable it is w.r.t to some of the hyperparameters.**
>
> A4: Hyper-parameter $\lambda$ is a weighting parameter to trade-off $L_{HPCL}$ and $L_{DPCL}$. We conducted extensive experiments with various values of $\lambda$ (e.g., 0.1, 0.25, etc.) and found that $\lambda = 0.5$ provided the best results, as demonstrated in Figure 2a.
>
> **W5: Overall the discussion of the results is often a bit meaningless and just describes the values in the tables, partially even in a bit of a weird fashion.**
>
> A5: We acknowledge that the discussion of the results is insufficient. We admit there is still a large gap between our method and SOTA fully-supervised methods on SYSU-MM01. Nonetheless, fully supervised methods have drawbacks, such as high annotation costs. Additionally, our method can enhance the performance of popular methods like ADCA, PGM, and MMM. For instance, combining PCLHD with MMM improves its Rank-1 from 61.6% to 65.9% and mAP from 57.9% to 61.8%, and the gain is remarkable by 4.3% and 3.9%. The improvement indicates the benefits of focusing on divergence and variety in this field, and demonstrates that our approach is not only effective on its own but also beneficial when integrated with other methods.
>
> ### Questions
> **Q1: Why are the "Baseline" results in table 2 so different from the PGM results?**
>
> A1: The official paper may have contained typos, as we obtain better results on indoor search using the official code. The better performance on indoor search is normal because indoor environments are relatively simple, stable lighting and fewer background distractions, making matching easier. Most methods in Table 1 also show better results for indoor search compared to all search.
>
> **Q2: Why did you duplicate almost every single equation just for visible and infrared?**
>
> A2: Thanks for your valuable comments, we will revise this in the future version.
>
> **Q3: In figure 2c, it very much confused me to see a line between the data points since the different methods are in no way connected.**
>
> A3: Thank you for your suggestion, we will replace it with a bar chart to improve clarity.
>
> **Q4: Is figure 3 generated based on training of test images?**
>
> A4: The t-SNE map in Fig.3 is plotted by 10 randomly chosen identities of test images on SYSU-MM01.
>
> ### Limitations
> **L1: I find it somewhat lazy to not put some effort into coming up with a meaningful discussion of what still needs to be improved.**
>
> A1: Our method relies on the DBSCAN to generate pseudo-labels. For extremely large-scale datasets, DBSCAN might not perform well, thereby affecting the overall method's performance. Besides, if the data has a large intra-class discrepancy, it might impact the results, and we plan to address this in future research by hierarchical clustering.
>
> **L2: This is actually outrageous: "There is no societal impact of the work performed."**
>
> A2: Thank you for your thorough review and detailed feedback. We apologize for the misunderstanding. As you rightly pointed out, performing this kind of research can significantly impact the effectiveness of surveillance. We mistakenly interpreted the Broader Impacts section as referring only to negative societal impacts, hence our response of [NA]. We will review and correct the Paper Checklist accordingly.

---

> > ### Comment · Reviewer_edeS · 2024-08-09
> >
> > Could you actually elaborate on W3? You plan to add additional related work, but maybe give a list of the papers you plan to discuss here? As is, I can only hope you will come up with a meaningful discussion and that doesn't feel very convincing.
> >
> > Also, as another reviewer highlighted, will you be adding a limitations section to the paper?
> >
> > You have promised to add quite some things to the final paper and I'm wondering how all of this will fit and which parts will be sacrificed and moved to the supplementary to make all of this fit.
> >
> > Finally, in another comment you wrote "For instance, in image-text matching tasks, hard and dynamic prototypes help capture divergence and variety in visual and textual data, leading to better alignment between the two modalities." Is this some idea that you came up with, or is there an actual paper that does this?

---

> ### Author Response · Authors · 2024-08-10
>
> We sincerely thank you once again for your valuable comments. We have carefully considered your concerns and have provided our responses below.
>
> **C1: Could you actually elaborate on W3? You plan to add additional related work, but maybe give a list of the papers you plan to discuss here? As is, I can only hope you will come up with a meaningful discussion and that doesn't feel very convincing.**
>
> A1: Thank you for your follow-up question. We plan to additional related work about unsupervised learning in general, specifically with a focus on contrastive learning and its modifications. Specifically, we will discuss the following related works:\
> Contrastive learning has achieved remarkable advancements in visual representation learning by pulling positive samples closer together and pushing negative samples further apart in feature space[1,2,6]. In contrastive learning, positive and negative samples play a crucial role and have a direct impact on the model's performance. Therefore, some data augmentation methods[2,3] have been proposed to augment positive pairs to learn discriminative features. For negative pairs, which have received less attention in contrastive learning, MoCo[1,4,5] series construct a memory bank to maintain numerous negative samples for increasing model’s performance. However, these methods define samples from different instances as negative, which is not feasible for ReID. In ReID datasets, many samples have the same identity, causing many positive samples to be wrongly labeled as negative.\
> [1]Kaiming He, Haoqi Fan, Yuxin Wu, Saining Xie, and Ross Girshick. Momentum contrast for unsupervised visual representation learning. CVPR, 2020.\
> [2]Xuyang Zhao, Tianqi Du, Yisen Wang, Jun Yao, and Weiran Huang. ArCL: enhancing contrastive learning with augmentation-robust representations. ICLR, 2023.\
> [3]Demirel B U, Holz C. Finding order in chaos: A novel data augmentation method for time series in contrastive learning. NeurIPS, 2023.\
> [4]Xinlei Chen, Haoqi Fan, Ross Girshick, and Kaiming He. Improved baselines with momentum contrastive learning. arXiv preprint, 2020.\
> [5]Xinlei Chen, Saining Xie, and Kaiming He. An empirical study of training self-supervised vision transformers. ICCV, 2021.\
> [6]Jinyu Yang, Jiali Duan, Son Tran, Yi Xu, Sampath Chanda, Liqun Chen, Belinda Zeng, Trishul Chilimbi. Vision-language pre-training with triple contrastive learning. CVPR, 2022. \
> **If you have any more relevant works in mind, we would greatly appreciate your suggestions.**
>
> **C2: Also, as another reviewer highlighted, will you be adding a limitations section to the paper?**\
> A2: We greatly value the reviewers' feedback and will be adding a limitation section as shown in the general response to the manuscript. We believe this addition will provide a more complete understanding of our work.
>
> **C3: You have promised to add quite some things to the final paper and I'm wondering how all of this will fit and which parts will be sacrificed and moved to the supplementary to make all of this fit.**\
> A3:To accommodate the additional content, we plan to move the discussion of Unsupervised Single-Modality Person ReID in the related work section, as well as Figure 2a and its related discussion, to the supplementary material. This will allow us to maintain the core content of the paper while ensuring that all important details are still accessible.
>
> **C4: In another comment you wrote "For instance, in image-text matching tasks, hard and dynamic prototypes help capture divergence and variety in visual and textual data, leading to better alignment between the two modalities." Is this some idea that you came up with, or is there an actual paper that does this?**\
> A4: This idea, to the best of our knowledge, has not appeared in any existing papers. It is an original concept that we plan to explore further as we expand this work.

---

> ### Author Response · Authors · 2024-08-13
>
> Dear Reviewer edeS,
>
> We sincerely appreciate your insightful review, as it plays a pivotal role in enhancing the quality of our manuscript. Furthermore, we have carefully considered your feedback to refine our paper further. As the discussion phase comes to an end, we want to ensure that we have thoroughly addressed all of your concerns. If there are any additional points of concern that you believe would contribute to the improvement of our manuscript, please let us know. We genuinely appreciate your expertise and the dedicated time you have devoted to reviewing our work.
>
> Warmest regards,
>
> Authors

---

### Author Rebuttal · Authors · 2024-08-06

We thank all the reviewers for their careful reading of our paper and help with improving our manuscript. We sincerely appreciate that you find our work:

- sense idea (edeS)

- well-writting (mgXA, i5Cg, n3oM)

- coherent and engaging (mgXA, i5Cg)

- innovative idea and fresh perspective (mgXA)

- clear motivation (mgXA, i5Cg, n3oM)

- higher-quality pseudo labels (i5Cg, n3oM)

- strong performance (edeS, mgXA, i5Cg)



We list the replies to several main concerns and suggestions raised by reviewers:

- **Question on the Unsupervised Setting of VI-ReID (Reviewer edeS and mgXA).**  We would like to clarify that we did not use the ground-truth labels and the number of classes. We followed the same experimental setup as existing unsupervised VI-ReID methods such as ADCA, PGM, and GUR. We used labels and the number of clusters generated by the DBSCAN clustering algorithm. It is a general setting and does not violate the unsupervised setting. We will clarify this in the final version.

- **Missing Discussion on Limitations (Reviewer edeS and mgXA).** Our method relies on the DBSCAN clustering algorithm to generate pseudo-labels. For extremely large-scale datasets, DBSCAN might not perform well, thereby affecting the overall method's performance. Besides, if the data has a large intra-class discrepancy, it might impact the results, and we plan to address this in future research by hierarchical clustering.

- **Missing Discussion on Different Baselines (Reviewer edeS and n3oM).** Most unsupervised VI-ReID methods overlook the aspects of divergence and variety. Our method significantly focuses on these aspects. Specifically, when combined with our PCLHD, ADCA's Rank-1 improves from 45.5% to 52.5%, and MMM's Rank-1 improves from 61.6% to 65.9%. This demonstrates the importance of divergence and variety.

We have conducted additional experiments. If not specified, we conduct analysis experiments on SYSU-MM01 in the single-shot &all-search mode.

**Additional experiments on ablation study of metrics for representing diversity (Reviewer mgXA).** For more details, please refer to Q1 for Reviewer mgXA.

**Additional experiments on ablation study of hyper-parameter $E_{CPCL}$ (Reviewer mgXA).** For more details, please refer to Q3 for Reviewer mgXA.

**Additional experiments in different baseline (Reviewer n3oM).** For more details, please refer to W1 for Reviewer n3oM.

---

### Decision · Program_Chairs · 2024-09-25

**Decision:**

Accept (poster)

**Comment:**

This paper proposes an unsupervised approach to Infrared-Visible person re-identification. The authors propose to use progressive contrastive learning with hard and dynamic prototypes. Hard prototypes are used to find divergent samples, while dynamic ones are used to conserve variability. Reviewers raised a number of technical issues, most of which the authors responded to convincingly (or at least partially) during the rebuttal period. On the positive side, reviewers praise the clarity of technical presentation and motivation, the excellent experimental results, and the innovative use of contrastive learning to enhance discrimination.

The two Ethics Reviewers and *all four reviewers* note that there is absolutely no discussion of limitations of the work and potential societal impacts. During the rebuttal period the authors demonstrated very little willingness to engage at all with the broader implications of surveillance technologies. Although the reviewers (and one Ethics Reviewer) note that the work stands on its own technical merits, this lack of engagement to the point of defensiveness is unacceptable. The authors are encouraged to seriously reflect on these issues in any final version of this work.